# Augmentation of 5-Aminolevulinic Acid Treatment of Glioblastoma by Adding Ciprofloxacin, Deferiprone, 5-Fluorouracil and Febuxostat: The CAALA Regimen

**DOI:** 10.3390/brainsci8120203

**Published:** 2018-11-22

**Authors:** Richard E. Kast, Nicolas Skuli, Iacopo Sardi, Felix Capanni, Martin Hessling, Guido Frosina, Anton P. Kast, Georg Karpel-Massler, Marc-Eric Halatsch

**Affiliations:** 1IIAIGC Study Center, 22 Church Street, Burlington, VT 05401, USA; antonkast@gmail.com; 2Abramson Family Cancer Research Institute, University of Pennsylvania, Cancer Center, 438 BRB II/III, 421 Curie Boulevard, Philadelphia, PA 19104-6160, USA; nicskuli@pennmedicine.upenn.edu; 3Neuro-Oncology Unit, Department of Pediatric Oncology, Meyer Children’s Hospital, viale Pieraccini, 24, 50139 Florence, Italy; iacopo.sardi@meyer.it; 4Rer. Hum. Biol, Department of Mechatronics and Medical Engineering, Ulm University of Applied Sciences, Albert-Einstein-Allee 55, D-89081 Ulm, Germany; capanni@hs-ulm.de; 5Department of Mechatronics and Medical Engineering, Ulm University of Applied Sciences, Albert-Einstein-Allee 55, D-89081 Ulm, Germany; hessling@hs-ulm.de; 6Mutagenesis & Cancer Prevention Unit, IRCCS Ospedale Policlinico San Martino, Largo Rosanna Benzi 10, 16132 Genova, Italy; guido.frosina@hsanmartino.it; 7Department of Neurosurgery, Ulm University Hospital, Albert-Einstein-Allee 23, D-89081 Ulm, Germany; georg.karpel@gmail.com (G.K.-M.); marc-eric.halatsch@uniklinik-ulm.de (M.-E.H.)

**Keywords:** 5-aminolevulinic acid, ciprofloxacin, deferiprone, fluorescence, 5-fluorouracil, febuxostat, glioblastoma, photodynamic treatment, temozolomide,

## Abstract

The CAALA (Complex Augmentation of ALA) regimen was developed with the goal of redressing some of the weaknesses of 5-aminolevulinic acid (5-ALA) use in glioblastoma treatment as it now stands. 5-ALA is approved for use prior to glioblastoma surgery to better demarcate tumor from brain tissue. 5-ALA is also used in intraoperative photodynamic treatment of glioblastoma by virtue of uptake of 5-ALA and its preferential conversion to protoporphyrin IX in glioblastoma cells. Protoporphyrin IX becomes cytotoxic after exposure to 410 nm or 635 nm light. CAALA uses four currently-marketed drugs—the antibiotic ciprofloxacin, the iron chelator deferiprone, the antimetabolite 5-FU, and the xanthine oxidase inhibitor febuxostat—that all have evidence of ability to both increase 5-ALA mediated intraoperative glioblastoma demarcation and photodynamic cytotoxicity of in situ glioblastoma cells. Data from testing the full CAALA on living minipigs xenotransplanted with human glioblastoma cells will determine safety and potential for benefit in advancing CAALA to a clinical trial.

## Highlights

Oral 5-aminolevulinic acid is preferentially converted to intracellular protoporphyrin IX in glioblastoma cellsThis allows intraoperative fluorescence assisted resection and photodynamic treatmentCAALA uses four repurposed drugs to increase glioblastoma-specific intracellular protoporphyrin IXIncreased protoporphyrin IX increases effectiveness of fluorescence assisted resection and photodynamic treatment

## 1. Introduction

The CAALA (Complex Augmentation of ALA) regimen was developed with the simple goal of redressing some of the weaknesses of 5-aminolevulinic acid (5-ALA, synonymous with delta-aminolevulinic acid, trade name Gliolan^TM^, IDT Biologika GmbH, Germany) use in glioblastoma (GB) treatment as it now stands [1,2,3]. Preoperative 5-ALA is widely used to better demarcate GB tissue from normal brain tissue by virtue of GB cells’ preferential uptake of 5-ALA and preferential conversion to protoporphyrin IX (PpIX) [1,2,3]. PpIX fluoresces at ~635 nm after illumination with ~410 nm, allowing closer to complete resection. 

5-ALA is also used in photodynamic treatment (PDT) [4]. In PDT preoperative 5-ALA is taken up and preferentially converted to PpIX in GB cells’ mitochondria. PpIX emits a photon that results in cytotoxicity after exposure to ~410 nm or ~635 nm light. Both uses rest on the relatively preferential uptake into GB cells and the intracellular conversion to PpIX, commonly found to be 50:1 compared to non-malignant glia [4,5,6]. 5-ALA PDT is an important addition to existing intraoperative methods to distinguish high-density GB tissue from surrounding brain [6]. 

Preferential uptake does not occur to the same degree with other clinically used fluorochromes like indocyanine green, methylene blue, and fluorescein compared to 5-ALA [6].

By the time GB is clinically identified, multiple microscopic islands exist within brain substance surrounding the main tumor mass [7,8]. A further and related problem is the migration-enhancing effect of surgery, cytotoxic chemotherapy and of irradiation. There does not seem to be a critical single invasion-related target, no single Achilles’ heel, to stop migration [9,10]. Therefore CAALA attempts to reach the GB micro-islands in the resection margin with a mechanical GB cell-preferential method.

CAALA regimen has been designed to use several heretofore clinically unexploited aspects of 5-ALA in redressing some of 5-ALA’s weaknesses. We will show how past data indicate that four already-marketed drugs could be added together to increase PpIX content of GB cells, increasing both 5-ALA fluorescence guided surgery and PDT.

Drug repurposing, as in the CAALA regimen, is the application of already-marketed drugs to treat diseases for which they were not originally indicated. Repurposed drugs have known safety profiles and can bypass much of the cost and time needed to bring new drugs to market. Since median survival after a GB diagnosis remains under two years, amazingly despite 600+ human clinical trials reporting in the last ~10 years, it is time to explore new avenues of approach. The CAALA Regimen is one such principally new avenue. 

The four 5-ALA augmentation drugs are ciprofloxacin, a broad-spectrum antibiotic; deferiprone, an iron chelating drug; 5-fluorouracil (5-FU), an antimetabolite in use to treat colon or breast cancer; and febuxostat, a xanthine oxidase inhibitor used in the treatment of gout. All have a reasonably robust preclinical database, indicating that they will, by various mechanisms, individually or potentially to even greater degree when used all together, increase 5-ALA’s effectiveness in both 5-ALA’s roles. 

5-ALA is a precursor used in physiological multistage PpIX synthesis. PpIX, in turn, is an essential precursor in physiological heme synthesis. PpIX is an endogenous fluorophore. After chelating ferrous ion (Fe++) mediated by ferrochelatase, PpIX becomes heme and no longer fluoresces, see Figure 1. PpIX becomes a cytotoxic reactive oxygen species (ROS) generating photophore after light excitation at ~410 nm (deep violet) or ~635 nm (red) light. PpIX also fluoresces at ~635 nm (red) after ~410 nm excitation, thereby indicating where large accumulations of GB cells reside that might otherwise be missed by the surgeon [2,3,7,11].

Typical 5-ALA dosing in GB would be 20 mg/kg p.o. 3 h prior to anesthesia. Side effects from this single bolus dose or during standard clinical PDT in GB tend to be limited to transient liver function enzyme elevations, and minor skin stinging if exposed to sunlight [12]. Serious side effects are remarkably absent for such a helpful medicine. In study of 50 mg oral 5-ALA/day given over 12 weeks to prediabetic patients, no side effects were seen [12]. In a dose escalation study of 5-ALA in GB surgery using 50 mg/kg as opposed to the standard 20 mg/kg, Cozzens et al found only transient grade 1 skin redness and peeling [13]. Skin sensitivity to light (pain) can occur with standard 5-ALA dosing of 20 mg/kg per treatment session, so protection from sunlight is usual.

## 2. Intraoperative Fluorescence Tumor Demarcation

Intraoperative PpIX fluorescence-guided resection gives longer progression-free survival. Two studies from 2017 showed preferential overall survival with 5-ALA guided resection [14,15]. 

Intraoperative 5-ALA fluorescent demarcation of GB tissue is in several ways imperfect [16,17,18,19]. Crucially, for partly unidentified reasons, some areas within a GB fail to stain. Intraoperative MRI areas of enhancement are not perfectly coincident with 5-ALA fluorescence areas and vice versa. Modalities do not completely overlap, with MRI and 5-ALA each identifying GB tissue the other misses. 

Cavity margins after 5-ALA assisted resection even when clear of any fluorescence on surgical microscope examination tend to have isolated GB nests or single cells on microscopy, some of which fluoresce weakly, some of which do not fluoresce [20,21,22,23,24].

In parallel and related to above, 5-ALA PDT cytotoxicity is not as thorough as we would like it to be. In CAALA we aim to redress these weaknesses by the combined action of the four drugs resulting in increased intracellular PpIX.

As evident by clinical behavior and the above considerations, the term “tumor margin” as used in GB is a misnomer. By the time a GB is clinically recognized, small microscopic islands or even single GB cells have spread deep into brain tissue. There is no true tumor margin. So “gross total resection” refers to complete resection of enhancing tumor on MRI and complete removal of post-5-ALA fluorescence under the low power surgical operating microscope. It is essentially never complete. 

Of interest and relevance to the rationale driving CAALA, low-grade gliomas that do not overtly fluoresce with intraoperative 5-ALA microscopy nevertheless, do contain slightly disproportionately excess intracellular PpIX compared to normal glia after preoperative 5-ALA [25]. 

This and other observations suggest PpIX accumulates proportionally to malignancy degree in that more aggressive GBs tend to give stronger fluorescence [23,26,27]. The area of MRI enhancement tends to correspond to areas of fluorescence. Fluorescence under 410 nm light after 5-ALA, 20 mg/kg orally, given 3 h preoperatively was 100% predictive of high-grade glioma [26,27].

Additionally, in accord with what we know about the spatial heterogeneity of receptor kinases and other markers, PpIX fluorescence is spatially heterogeneous both area wise within the main GB tumor body, and in individual GB cells. In GB areas of high ~635 nm fluorescence, there are individual cells that fail to fluoresce and in GB areas that do not overtly fluoresce there are individual cells that do [27].

Nakayama et al. showed good 5-ALA uptake and conversion to PpIX in dormant, non-cycling prostate cancer cells [28]. It will be important to determine if this is the case in GB.

5-ALA has a circulating Tmax at 1 h post oral ingestion of 20 mg/kg, with a T_1/2_ of 3 h. Circulating PpIX peaks at 8 h.

5-ALA use in glioma surgery was reviewed twice in 2017 [15,29]. Four papers published in 2018 recounted the various factors active in determining the degree of GB cells’ PpIX content [2,3,4,5]. Since 100% of strongly fluorescing tissue during 5-ALA assisted surgery was clearly GB tissue but 49% of non-overtly fluorescing resection margin tissue also contained a few GB cells or cell islands, if we can indeed increase 5-ALA uptake and conversion to PpIX as aimed for with CAALA, we might be able to catch microfoci of GB tissue in PDT that would otherwise be missed [30]: The fundamental idea behind CAALA.

CAALA augmentation paths fall into four categories:
Increase GB cell uptake of 5-ALA by increasing oral doseIncrease conversion of 5-ALA to PpIX by using 5-FU and ciprofloxacinDecrease GB cell efflux of PpIX with febuxostatDecrease further metabolism of PpIX to non-fluorescent, PDT inactive heme, with deferiprone


## 3. The Drug

See Figure 1 for a schematic of the points of expected action for the four CAALA drugs.

### 3.1. Ciprofloxacin

The fluorinated quinolone ciprofloxacin is a 331 Da broad-spectrum antibiotic with low protein binding allowing good brain tissue levels. 

Empirically, the physiologic concentration of ciprofloxacin enhances 5-ALA cytotoxicity in vitro via enhanced conversion of 5-ALA into PpIX, discovered in a general chemical screen [31]. This was confirmed later with a demonstration of ciprofloxacin enhanced 5-ALA cytotoxicity to meningioma cells [32] and to glioma cells by the same mechanism where increased ROS in response to KeV radiation could also be shown [33]. 

Concordant with these data, ciprofloxacin treated rats had reduced brain glutathione and catalase levels [34]. Ciprofloxacin in supraphysiologic doses increased brain oxidative stress in rats [35] and thus would increase the effect of photoexcited PpIX ROS generation if the rat data were replicated in humans at attainable clinical doses.

### 3.2. Deferiprone

There are currently three marketed pharmaceutical iron-binding drugs: deferiprone, deferoxamine, and deferasirox. Deferiprone is a 139 Da iron binding drug, approved for use in iron overload states as in thalassemias and other damaging iron accumulation states. It is the only one that has good brain tissue levels and documented lowering of brain iron.

Any decrease in activity or amount of ferrochelatase leads to intracellular accumulation of PpIX by inhibiting ferrochelatase’s conversion of PpIX to inactive heme, as depicted in Figure 1 [36,37]. GB cells tend to have lower amounts of ferrochelatase than corresponding normal glia, thus contributing an element (among others) to GB cell-preferential fluorescence after 5-ALA and consequent to that, preferential cytotoxicity with PDT [37]. Indeed a deficiency of ferrochelatase activity may be the primary reason GB cells preferentially overgenerate PpIX after 5-ALA exposure compared to normal glia and neurons given the linear direct relationship found between intracellular ferrochelatase and PpIX after 5-ALA [38].

Silencing of ferrochelatase results in up to a 50-fold increase in intracellular PpIX accumulation in several cancer cell lines [39]. Unique among pharmaceutical iron chelators, deferiprone gets good brain tissue levels and lowers brain iron accumulations [40,41].

Interestingly, deferiprone itself, without 5-ALA or any PDT, augmented temozolomide cytotoxicity to glioma cells [42]. 

Glioma cells with stem cell attributes take up less 5-ALA and make less PpIX, have weaker fluorescence, and grow faster when orthotopically xenografted, than the main GB cell population. Exposure of these glioma stem cells to the iron chelator deferoxamine resulted in increased PpIX and fluorescence to equal that of non-stem cells [43].

Anand et al., in 2012 reviewed and advocated iron chelation, 5-FU, methotrexate and vitamin D as potential 5-ALA PDT augmentation paths [44]. We echo this principle, particularly for iron chelation, and update the implied suggestion of Anand et al., expanding on this theme with the additions of CAALA-increased 5-ALA dose, ciprofloxacin, and febuxostat.

### 3.3. 5-Fluorouracil (5-FU)

5-FU is a 130 Da, very short half-life, non-protein bound anti-metabolite used in cancer treatment. Capecitabine is an FDA, EMA, and Health Canada approved 359 Da oral pharmaceutical precursor to 5-FU used as a convenient way to deliver 5-FU when treating colon, breast, and other cancers. Capecitabine metabolically converts into 5-FU after ingestion. It would be capecitabine that we envision using to deliver 5-FU during CAALA, keeping the regimen entirely oral. Overall survival in GB treated without PDT, but with capecitabine 625 mg/m**^2^** given twice daily during 6-week radiation and the following month, thereafter at 14 days on, 7 days off, was ~1 year [45]. Significantly, this was below current mean with standard post-resection temozolomide and irradiation.

5-FU enhances 5-ALA photocytotoxicity. In a study of rats with UV induced squamous cell carcinoma, sequential 3 days of 5-FU followed by 4 h of 5-ALA gave a 2- to 4-fold increase of intracellular PpIX compared to rats not given 5-FU pretreatment [46]. Importantly, normal non-irradiated skin showed no difference in intracellular PpIX with either sequential 5-FU plus 5-ALA or when given 5-ALA alone compared to skin cells exposed to neither [46]. Human clinical use in using sequential topical 5-FU, followed by 5-ALA PDT compared with 5-ALA PDT alone in treating actinic keratoses indicate that pretreatment with 5-FU increases 5-ALA PDT effectiveness [47,48].

### 3.4. Febuxostat 

Febuxostat is a 316 Da xanthine oxidase inhibitor used to treat gout [49].

Primary cell efflux of PpIX is affected by the ABCG2 exporter pump [50]. Approved and marketed to treat hyperuricemia, febuxostat is a potent inhibitor of ABCG2, IC50 being lower than its usual plasma levels [51]. We may expect to increase GB intracellular PpIX by inhibiting its efflux, enhancing both PDT and 5-ALA assisted resection. 

Xenobiotics generally are exported by ABCG2 (same as the breast cancer resistance protein, BCRP) as well as several other efflux transporters. Of particular relevance to GB, temozolomide [52], erlotinib [53] and afatinib [54] brain tissue levels are many folds higher when ABCG2 is inhibited or knocked down, as we intend to achieve with febuxostat.

## 4. Cautions and Discussion

We present a potential new treatment for a fatal disease with few treatment options. CAALA is a low-risk amalgam of currently used 5-ALA with four well tolerated common non-oncology drugs that have evidence of enhancing 5-ALA benefits. CAALA holds promise to improve outcomes by enabling a more effective intraoperative and postoperative 5-ALA mediated PDT and current standard chemoirradiation. 

The temporal relationship between preferential 5-ALA uptake by glioblastoma cells, within minutes [55], and the later development, within hours [56], of maximal fluorescence—for which PpIX is responsible—suggests that both greater avidity of 5-ALA and stronger weighting of PpIX synthesis are responsible for glioblastoma fluorescence. As discussed above, and evident in the summary Figure 1, CAALA drugs act primarily by enhancing PpIX formation from intracellular 5-ALA and inhibiting further processing or export of PpIX. It is by these attributes that preferential glioblastoma cell cytotoxicity is achieved with CAALA.

Dexamethasone, desipramine, levetiracetam, phenytoin, and valproic acid are drugs that tend to diminish GB fluorescence after 5-ALA so might be best avoided during 5-ALA PDT and CAALA [57,58]. In addition, potential generators of increased nitric oxide tend to antagonize 5-ALA [59] and so should be avoided.

Potential for growth stimulation: Low dose ROS generation by PpIX can be proliferation stimulating where higher doses are cytotoxic [60]. CAALA will, therefore, be in accord with the precept of Niccolo Machiavelli (1469–1527), “If an injury has to be done to a man it should be so severe that his vengeance need not be feared”. Any tissue disruption-surgery, cytotoxic drugs, irradiation-tends to trigger epithelial-to-mesenchymal transition (EMT). In EMT sessile GB cells acquire a mobile, mesenchymal phenotype, becoming detached from the main tumor mass [61,62,63]. Thus the 5-ALA PDT can be expected to deliver an EMT signal to any GB cells that are harmed but not killed, as Machiavelli cautioned. CAALA was designed to lower GB’s vengeance potential.

Preclinical in vitro testing to document additivity (or not) of the four drugs will determine what drug mix goes forward to a minipig GB xenograft 5-ALA PDT study preparatory to a potential clinical phase 1 trial.

The blood–brain barrier, BBB, will potentially to some degree limit exposure to the ancillary CAALA drugs. Deferiprone passes the BBB well and has good brain tissue levels [40,41]. Ciprofloxacin does enter the brain but to a lesser extent [64,65] and 5-FU to an even lesser extent [66]. Febuxostat’s ability to cross the BBB has not been studied, but side effects of nausea and headache would suggest some does get through. The BBB is compromised in and around a glioblastoma so we can expect greater CAALA drug brain tissue levels there, but achieving adequate brain tissue levels might require dedicated BBB opening maneuvers.

Crucial for the preclinical testing phase will be determining if any of the proposed CAALA drugs decrease GB cell selectivity of 5-ALA uptake and conversion to PpIX. If any do reduce selectivity of uptake to a significant degree in preclinical testing, they will be dropped from the regimen.

## 5. Conclusions

Although some risk attends a polypharmacy, multimodal regimen like CAALA, we believe the gravity of a GB diagnosis warrants proceeding with in vitro and minipig testing followed by clinical trial should the animal study reveal no unexpected toxicities. 

1. We assembled evidence that four common well tolerated drugs (1) ciprofloxacin, (2) deferiprone, (3) 5-FU and (4) febuxostat-will augment both 5-ALA’s roles in treating GB. All four drugs have documented mechanisms of action and preclinical potential to increase the PpIX content of GB cells, (a) thereby increasing intraoperative fluorescence identification of GB tissue, and (b) increasing intraoperative PDT effectiveness.

2. Higher 5-ALA doses might be safely given with consequent potential for increased intracellular PpIX.

3. A consensus has formed within the GB research community that effective treatment must target multiple active growth pathways that will, in turn, require a multi-drug, multi-modal approach for cure or long-term growth suppression. In accord with that view, the CAALA Regimen is a potential addition to the panoply of approaches currently available [67].

## Figures and Tables

**Figure 1 brainsci-08-00203-f001:**
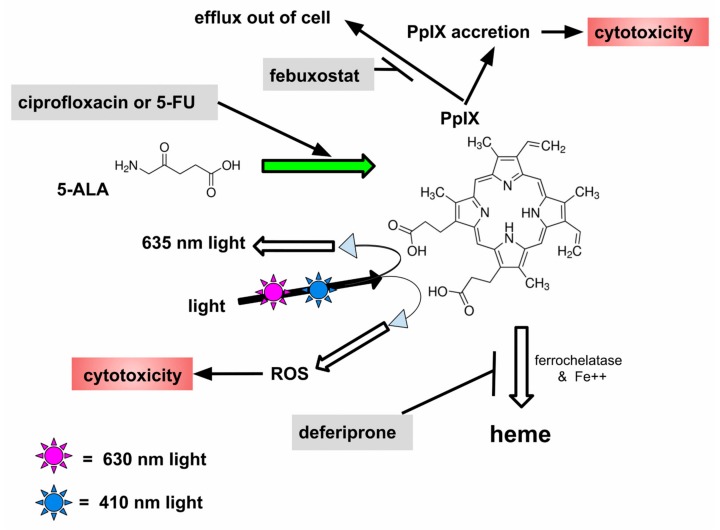
Schematic of pharmaceutical elements of our proposal, many intermediate steps are not depicted. The text gives specifics of how ciprofloxacin and 5-FU might increase 5-ALA uptake and PpIX synthesis inside GB(glioblastoma) cells and explains how deferiprone inhibits loss of PpIX by inhibiting its conversion into heme and efflux inhibition by febuxostat.

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
