# Peer review of "Augmentation of 5-Aminolevulinic Acid Treatment of Glioblastoma by Adding Ciprofloxacin, Deferiprone, 5-Fluorouracil and Febuxostat: The CAALA Regimen"

_brainsci, 2018, doi:10.3390/brainsci8120203_

Round 1
Reviewer 1 Report
This concept paper summarizes a protocol for the augmentation of two features of 5-ALA: its ability to highlight glioblastoma cells and its ability to be phototoxic to those same cells. The authors comment on the variability of the 5-ALA-PpIX metabolic pathway which results in its variable illumination of glioblastoma cells, making it less than ideal as a marker for fluorescence guided surgical resection and as a photodynamic therapy (PDT). In this manuscript, they postulate that the addition of a cocktail of drugs that all impact 5-ALA or PpIX pathways will significantly improve the utility of 5-ALA in glioblastoma resection and PDT.
This concept is a very intriguing one, and has been clearly thought out from the perspective of capitalizing on drugs that are already approved by the US FDA and regulatory boards in other jurisdictions. It is less clear how the drugs will specifically target the glioblastoma cells rather than surrounding normal cells and whether all of the drugs are blood brain barrier permeable. While theoretically this CAALA drug cocktail is good idea, further details are required.
1. With regards to BBB permeability, only deferiprone appears to have been demonstrated as permeable, as suggested on line 182, although without any reference. Can the authors support that the other drugs would also be able to permeate the BBB? In some instances this seems implied, as with the assertion that temozolimide, erlotinib and afatinib are higher in brain when ABCG2 is inhibited or knocked-down, but this does not mean that the ABCG2 inhibitor febuxostat is BBB permeable. Please state with references when this is the case for all the proposed drugs.
2. Without targeting drugs specifically to glioblastoma cells, it seems that there may be deleterious systemic toxicity to other cells in the body as well as the brain. For example, ciprofloxacin “increased brain oxidative stress” (line 175), is used in support of its ability to augment the cytotoxicity of PpIX ROS, but without targeting it, wouldn’t this stress be inflicted on all cells? The side-effects could be alleviated by controlling the dose. What doses of the drugs are recommended for CAALA that reduce potential off-target side effects?
3. Similarly, would the reduction of ferrochelatase by the addition of deferiprone result in the augmentation of PpIX fluorescence in all cells, not only glioma cells?
4. At least one reference is not listing the correct citation: line 137 cites ref. 25, Stummer et al., 2017, for support of the assertion that low grade gliomas do not fluoresce with 5-ALA yet contain “slightly disproportionately excess intracellular PpIX...” This is a relevant statement, but I could not find any supporting evidence in that reference.
Minor issues:
1. The word “selective” is used on lines 40, 64, 69, and 73. 5-ALA uptake in GBM cells is not selective; it occurs “preferentially” but not selectively.
2. Some statements are asserted without references; for example, lines 138-140, have several assertions that are not supported by references including “This, and other observations, suggest PpIX accretion…”
3. Line 78: delete “was can block or stimulate” to improve sentence clarity
4. Line 83: replace “benefitting thereby” with “improving”
5. Line 122: change to read “do not completely overlap”
6. Paragraph from 135-137 needs its reference amended.
7. Line 138 – "accretion proportionality" should probably be “accumulates proportionally”
8. Line 215, this statement is very important, so I would remove the “Perhaps even more” and just leave as “Importantly”
Author Response
Responses to Reviewer 1. Re. point #1: “ It is less clear how the drugs will specifically target the glioblastoma cells rather than surrounding normal cells…” We address this interesting matter by adding… “The temporal relationship between preferential 5-ALA uptake by glioblastoma cells, within minutes (D), and the later development, within hours (E), of maximal fluorescence - for which PpIX is responsible - suggests that both greater avidity of 5-ALA and stronger weighting of PpIX synthesis are responsible for glioblastoma fluorescence. As discussed above and evident in the summary Figure 1., CAALA drugs act primarily by enhancing PpIX formation from intracellular 5-ALA and inhibiting further processing or export of PpIX. It is by these attributes that selective glioblastoma cell cytotoxicity is achieved with CAALA.” Re. …”whether all of the drugs are blood brain barrier permeable. “... We address this by adding… “The blood-brain barrier, BBB, will potentially, to some degree limit exposure to the ancillary CAALA drugs. Deferiprone passes the BBB well and has good brain tissue levels [40, 41], ciprofloxacin enters the brain to lesser extent [A, B] and 5-FU to even lesser extent [C]. Febuxostat’s ability to cross the BBB has not been studied, but side effects of nausea and headache would suggest some does get through. The BBB is compromised in and around a glioblastoma so we can expect greater CAALA drug brain tissue levels there, but achieving adequate brain tissue levels might require dedicated BBB opening maneuvers.” Re point #2: Yes, ciprofloxacin does generally increase oxidative stress throughout the body. Also we don’t suggest that glioblastoma is any more likely to take up cipro than any other area of brain. But a] studies of cipro in various cancers indicate that cancer cells generally might be somewhat more sensitive to oxidative stress, and b] cipro is one of the most widely used antibiotics worldwide, and is without evidence of any common tissue destructive side effects. Re. point #3: Yes, inhibiting ferrochelatase with deferiprone will increase PpIX in all cells that have PpIX. But since, as we point out throughout the article, glioblastoma cells have greater avidity for 5-ALA than other brain cells, we expect the relative increase of PpIX in glioblastoma cells to be disproportionately amplified. Our reference 43 gives evidence to this effect. Re. point #4: Reference 25.seems correct to us. It is not to Stummer et al, it is to Valdes et al. Reference 24 is to Stummer et al, and on checking this seems correct too. Reference 23 reenforces this statement and has therefore been added at this point. Re. minor issues: #1. All uses of selective have been corrected to preferential. #2. Per reviewer’s suggestion, sentence amended to “This, and other observations suggest PpIX accretion proportionality to malignancy degree in that more aggressive GBs tend to give stronger fluorescence [23, 26, 27].” #3. Per reviewer’s suggestion, sentence amended to “There doesn’t seem to be a critical single invasion-related target, no single Achilles’ heel to stop migration [9, 10]. “ #4. Per reviewer’s suggestion sentence amended to “We will show how past data indicates that four already-marketed drugs could be added together to increase PpIX content of GB cells, increasing both 5-ALA fluorescence guided surgery and PDT.“ #5. Per reviewer’s suggestion sentence amended to “Modalities do not completely overlap - MRI and 5-ALA each can identify GB tissue the other misses. #6. Per reviewer’s suggestion sentence now reads “Of interest and relevance to the rationale driving CAALA, low grade gliomas that do not overtly fluoresce with intraoperative 5-ALA microscopy nevertheless do contain slightly disproportionately excess intracellular PpIX compared to normal glia after preoperative 5-ALA [25]. “ #7. Per reviewer’s suggestion sentence now reads “This, and other observations suggest PpIX accumulates proportionally to malignancy degree in that more aggressive GBs tend to give stronger fluorescence [23, 26, 27]. “ #8. Per reviewer’s suggestion sentence now reads “ Importantly, normal non-irradiated skin showed no difference in intracellular PpIX with either sequential 5-FU plus 5-ALA or when given 5-ALA alone compared to skin cells exposed to neither [46].”
Reviewer 2 Report
A few comments for the authors’ consideration:
1. Is there any known drug-drug interaction between these four 5-ALA augmentation drugs?
2. A table summarizing these four drugs in terms of their role in the CAALA, PK, brain-penetration, MOA, etc., might be helpful.
3. More detail about the mini pig model should be discussed? Why is this model suggested?
Author Response
Thank you for these helpful thoughts. Re. #1. There is no recognized drug-drug interaction between the CAALA drugs. Re. #2, the paper is long enough and we think an added Table would just complicate an already difficult paper. Would not the Figure serve a similar function as the Table you suggest ? We have added a short sentence on what little is known on brain penetration of the four augmentation drugs. Re. #3,The minipig model was chosen due to closer size to human brain. PDT equipment we intend to use will be standard clinically currently used instrument.
Reviewer 3 Report
the authors have written a hypothetical Concept paper wherein 4 relatively safe Drugs with specifically known mechanisms of action, have the affect of increasing concentrations of PpIX which in turn increases Florescence of glioblastoma cells and enhances the effect Of photodynamic therapy.
although the individual drugs have relatively safe therapeutic profiles, it is not certain how a combination of these drugs will act and individuals with glioblastoma's. Prior to developing clinical trials which most likely will be randomized, it is essential to get evidence in the lab using animal studies. This Concept paper is an excellent start to the process of developing greater animal-based projects which can then clinically translated from the bench to the bedside with randomized control trials. As the authors have noted, over the past 10 years and 600 or more clinical trials for these devastating tumors, there have not been any remarkable progress. As neuro oncologist we are looking for new paradigms to treat these tumors and this is a great start.
the authors could simplify some of the sentences and clarify them, decreased the use of Superlative in the sentences. For example in sentence #135the The authors begin with 'of interest and supreme relevance', which is unnecessary and they can replace that with a simple sentence such as; the rationale for the CAALA regimen and low-grade gliomas .....................
It is understood many of the citations and statements are significant in the Authors Do not need to overemphasize this.
the paper is otherwise relatively simple to understand , has no statistical methods.
Author Response
Yes, we agree and have accordingly deleted the superlatives and simplified the contorted constructions. Thanks for pointing this out.
Reviewer 4 Report
I have read with great interest the proposal of an in-vitro and in-vivo study on multi-drug regimen aimed to improve the effectiveness of 5-ALA in glioma surgery.
The concept seems valuable and I would be glad to read about its results in the near future.
However, even if a ethical comitee aproval is not required for a concept-paper like this, I would suggest authors to spend some lines to explain ethical motivations sustaining this in-vivo study.
Page 9, line 322: the supposed reference n.22 by Toms et al "Intraoperative optical 341 spectroscopy identifies infiltrating glioma margins with high sensitivity" is not numbered. Please correct it and pay attention to correctly refer it in the manuscript.
Author Response
Thank you for your supportive comments. We have amended wording to explain the ethical justification for study of CAALA. "
We present a potential new treatment for a fatal disease with few treatment options. CAALA is a low risk amalgam of currently used 5-ALA with four well tolerated common non-oncology drugs that have evidence of enhancing 5-ALA benefits. CAALA holds promise to improve outcomes by enabling a more effective intraoperative and post-operative 5-ALA mediated PDT and current standard chemoirradiation. "
Regarding your concerns about the Toms et al article " Intraoperative optical spectroscopy identifies infiltrating glioma margins with high sensitivity." We quote them here..."Cavity margins after 5-ALA assisted resection even when clear of any fluorescence on surgical microscope examination tend to have isolated GB nests or single cells on microscopy, some of which fluoresce weakly, some of which do not fluoresce [20-24]."